# The Food and Nutrition Environment at Secondary Schools in the Eastern Cape, South Africa as Reported by Learners

**DOI:** 10.3390/ijerph17114038

**Published:** 2020-06-05

**Authors:** Alice P. Okeyo, Eunice Seekoe, Anniza de Villiers, Mieke Faber, Johanna H. Nel, Nelia P. Steyn

**Affiliations:** 1Department of Nursing Science, University of Fort Hare, Ring Road, Alice 5701, South Africa; aokeyo@ufh.ac.za; 2Sefako Makgatho Health Science University, Ga-Rankuwa 0208, South Africa; Eunice.Seekoe@smu.ac.za; 3Research Capacity Development Division, South African Medical Research Council, Cape Town 7501, South Africa; Anniza.deVilliers@mrc.ac.za; 4Non-Communicable Diseases Research Unit, South African Medical Research Council, Cape Town 7501, South Africa; mieke.faber@mrc.ac.za; 5Centre of Excellence for Nutrition, North-West University, Potchefstroom 2520, South Africa; 6Department of Logistics, Stellenbosch University, Stellenbosch 7600, South Africa; jhnel@sun.ac.za; 7Division Human Nutrition, Department of Human Biology, University of Cape Town; UCT Medical campus, Anzio Road, Anatomy Building, Observatory, Cape Town 7925, South Africa

**Keywords:** obesity, adolescents, school environment, South Africa

## Abstract

Overweight and obesity are growing concerns in adolescents, particularly in females in South Africa. The aim of this study was to evaluate the food and nutrition environment in terms of government policy programs, nutrition education provided, and foods sold at secondary schools in the Eastern Cape province. Sixteen schools and grade 8–12 learners (*N* = 1360) were randomly selected from three health districts comprising poor disadvantaged communities. Based on age and sex specific body mass index (BMI) cut-off values, 13.3% of males and 5.5% of females were underweight, while 9.9% of males and 36.1% of females were overweight or obese. The main food items purchased at school were unhealthy energy-dense items such as fried flour dough balls, chocolates, candies, and crisps/chips. Nutrition knowledge scores based on the South African food-based dietary guidelines (FBDGs) were poor for 52% to 23.4% learners in Grades 8 to 12, respectively. Female learners generally had significantly higher nutrition knowledge scores compared to their male counterparts (*p* = 0.016). Questions poorly answered by more than 60% of learners, included the number of fruit and vegetable portions required daily, food to eat when overweight, foods containing fiber, and importance of legumes. It was noted that the majority of teachers who taught nutrition had no formal nutrition training and their responses to knowledge questions were poor indicating that they were not familiar with the FBDGs, which are part of the curriculum. Nutrition assessment as part of the *Integrated School Health Program* was done on few learners. Overall however, despite some challenges the government national school meal program provided meals daily to 96% of learners. In general, the school food and nutrition environment was not conducive for promoting healthy eating.

## 1. Introduction

The number of obese children and adolescents (aged five to 19 years) worldwide has increased from 11 million in 1975 to 124 million in 2016. If current trends continue, more children and adolescents will be obese than moderately or severely underweight by 2022 [1]. The high prevalence of overweight/obesity is of concern due to the many health problems associated with overweight and obesity in young people. Overweight and obesity are risk factors for non-communicable diseases (NCDs) including cardiovascular diseases, type 2 diabetes, certain cancers, and coronary heart disease [2]. In addition, obesity tracks from adolescence to adulthood [3]. Preventing adolescents’ obesity early in the life-course is therefore essential.

Adequate nutrition plays an important role in ensuring optimal growth and development in children and adolescents. Failure to meet the optimal nutritional requirements of an individual can lead to either undernutrition or obesity. The consequences of either undernutrition or obesity are deleterious to children’s and adolescents’ health and development [4]. Malnutrition undermines economic growth, perpetuates poverty and can make children vulnerable to abuse and exploitation [5]. Furthermore, nutrition also affects the cognitive development of learners and their school performance [6,7].

The 2016 South African Demographic and Health Survey (SADHS) indicated that 39.9% of females aged 15–24 years were overweight or obese and 5.8% were classified as thin. In males of the same age this was 11.2% and 15.8% respectively [8]. The Strategy for the Prevention and Control of Obesity 2015–2020 has a strong focus on preventing childhood obesity and aims to enable access to healthy food choices in various settings, including schools [9]. According to the strategy, nutrition education in schools should be in line with national recommendations. The food-based dietary guidelines (FBDGs) for South Africa were drafted in 2000, and officially adopted as national dietary guidelines in 2003 [10] and revised in 2013 [11]. A special manual comprising information on the FBDGs was developed by the Department of Basic Education and made available to life orientation (LO) teachers at schools [12]. Life orientation is a subject that covers basic important issues regarding health such as nutrition, hygiene, and physical activity. The subject is taught by general teachers and not nutrition educators. In addition, nutrition education and assessment are included in the Integrated School Health Program (ISHP) which was introduced in 2012 in South Africa, with the aim of contributing to the improvement of the general health of school-going learners as well as the environmental conditions in schools [13].

Socioecological models provide a conceptual framework to study factors influencing the dietary practices of learners because of their emphasis on multilevel linkages, the relationships among the multiple factors that impact on health and nutrition, and the focus on the connections between people and their environments [14]. The school food environment can have a pronounced impact on a child’s nutrition [15], since learners spend a substantial amount of time at school. For example, in South Africa, secondary learners spend on average 35 h a week in a school environment over four terms a year with breaks of two to four weeks between terms and they consume at least one meal a day at school; these meals are either from the school store (tuckshop), vendors or from meals provided by the National School Nutrition Program (NSNP), and/or learners’ lunch boxes from home. Thus, the school food environment is an important component in effective school-based interventions to promote healthy eating [15]. Numerous studies have indicated that many learners do not eat breakfast before coming to school and was one of the drivers of the government school meal program, the NSNP [16].

The ISHP provides a policy framework for adequate school environments and includes three school health packages and services, namely: health assessment and screening; health education and promotion; and on-site services by school nurses.

Thus, through the provision of onsite services, the ISHP addresses environmental issues related to food safety and sustainability as well as the school food and nutrition environment, in order to promote healthy eating practices. However, an empirical evaluation of the ISHP to provide evidence for policymakers concerning its effectiveness and implementation, particularly as it pertains to the dietary behaviors and weight status of secondary school learners, has rarely been conducted. The aim of the study was hence to evaluate the food and nutrition environment within secondary schools in urban and rural areas in the Eastern Cape in order to assess how healthy this environment is for learners.

## 2. Materials and Methods

### 2.1. Setting and Sample

The study was conducted in Eastern Cape province (Figure 1), which is in the south–eastern seaboard of South Africa. Eastern Cape is the second largest province in South Africa and had a total population of seven million people in 2017 [17]; the province had a total of 5589 public schools and a total learner population of 1,157,901 and 633,910 in primary and secondary schools, respectively [17]. The study covered three districts of the province, namely: Buffalo City Metropolitan, Chris Hani, and OR Tambo. The population for this study consisted of learners attending public secondary schools in quintiles 1, 2, and 3, and LO teachers working at the sampled schools. The quintile system is a ranking structure used by the government of South Africa which groups schools according to the poverty level of the community where the school is located. Schools in quintile 1 and 2 represent the poorest and all school funds come from the government. For the purpose of this study, schools from quintile 1 and 2 were grouped together, while quintile 3 formed another group of slightly higher socioeconomic status. All schools in quintiles 1–3 receive the NSNP meals.

A probability, multi-stage, cluster sampling method was used [18]. Three health districts (Buffalo City, OR Tambo, Chris Hani) of the eight district municipalities in the Eastern Cape province were purposively selected: two rural and one metropolitan (which includes urban and peri-urban regions, classified as urban). The three districts were chosen to allow for representation from socioeconomic clusters, urban and rural, and comparison of diverse contexts which provided a comprehensive picture of the school food and nutrition environment. From each of the three health districts, two education districts were randomly selected. Then, from each of these six selected educational districts, eligible secondary schools were randomly selected using a computer to include a sample of 16 schools [19].

In each of the selected grades (8–12) in the school, learners were selected using simple random balloting, thereby providing all learners an equal chance to participate in the study. Initially learners were asked to pick a “Yes” and “No” paper in a randomized manner. Those who had picked “Yes” were given a consent form to be signed by a parent/guardian. Only the learners who returned the signed form could sign the assent form to participate. A total of 1500 learners who provided consent participated in the study. However, due to incomplete data (questionnaires and/or anthropometry) for 140 participants, 1360 participants (528 males and 832 females), were included in the final sample and their data were used in the statistical analysis.

Data related to the school environment was also obtained from 18 LO teachers purposefully selected from those schools in the sample.

### 2.2. Selection and Training of Research Assistants

The study utilized five research assistants who received training during a workshop arranged for this purpose. The aim of the workshop was for the research assistants to become competent in all aspects of the research (i.e., anthropometric measurements, interviews, and questionnaire administration). The research assistants were final year Bachelor of Human Movement Science students in the Department of Human Movement Science, University of Fort Hare, as well as final year students of Food Technology at Walter Sisulu University.

### 2.3. Measurements

Overall, we studied those factors we believe contribute to the school food and nutrition environment and impact on the learner (Figure 2). These included foods provided by the NSNP, school stores and vendors selling food at schools, lunch boxes brought from home, and availability of vegetable gardens. We also collected information on breakfast eaten before school, as this could affect children’s school performance. We examined the nutrition education provided to learners at schools which included testing the knowledge of the learners and interviewing LO teachers at the schools. Another important aspect was to determine whether the ISHP was implemented at the schools, according to the learners.

### 2.4. Demographic Data on Learners

Information on the learner’s age, gender, ethnicity, grade, and their mother and father’s highest level of education was collected by questionnaire.

### 2.5. Anthropometry

Anthropometric measurements were taken with the learners in light clothing and no shoes, and in accordance with the International Society for the Advancement of Kinanthropometry (ISAK) recommendations [20]. Height was measured to the nearest 0.1 cm using a calibrated vertical stadiometer (Seca Portable 217 Seca, UK). Weight was measured to the nearest 0.01 kg using a calibrated digital electronic weighing scale (Seca 813, Seca, UK), which was calibrated after every 20th learner. Height and weight measurements were taken in duplicate and the average of the two measurements was calculated.

### 2.6. School Food and Nutrition Environment Information Supplied by Learners

Quantitative data on foods provided by the NSNP (number of meals received per week, number of vegetable and fruit servings per week), types of food bought from the school store and food vendors, types of food brought to school as lunch boxes, and the presence of a vegetable garden at school and its use was collected by questionnaire for each learner. Qualitative data on the learner’s view on the NSNP, school store, and food vendors were obtained from open-ended questions in the questionnaire.

In addition to the information provided by the learners, the research assistants made a note of the foods prepared and served on the day they visited the schools in order to obtain a list of the various food items served as part of the school meal, by NSNP, vendors, and school shops.

### 2.7. Nutrition Knowledge of Learners and Life Orientation Teachers

The nutrition knowledge of the learners was assessed using multiple choice and true/false questions, based on a validated 60-item questionnaire developed by Whati et al. [21] for South African adolescents aged 13 years to 20 years old. The nutrition knowledge questions were based on the recommendations of the FBDGs of South Africa. For the use of this study, the questionnaire was reduced to 40 questions (Appendix A). The shortened questionnaire had an overall Cronbach alpha value of 0.67, which is similar to what Whati et al. [21] found at disadvantaged schools. For each learner, the nutrition knowledge questionnaire was scored by giving a point for every correct answer, which were then summed to obtain the nutrition knowledge score out of a total of 40 points. The level of nutrition knowledge scores of the learners was divided into the following scores ≤16 (≤40%): poor; 17–24 (40%–63%): fair; ≥24 (≥63%): good.

All LO teachers in the sampled schools were interviewed by the research assistants using a structured questionnaire also containing open-ended questions to collect information on training in nutrition and knowledge of the FBDGs.

### 2.8. Implementation of the Integrated School Health Program

The ISHP provides a policy framework for adequate school environments and includes three school health packages and services, namely: health assessment and screening; health education and promotion; and on-site services. The questionnaire completed by the learners included questions on whether they had received the services of each of the three health packages, namely health assessment and screening; health education and promotion; and on-site services by school nurses.

### 2.9. Pilot Study

A pilot study was done to ascertain the logistical and technical procedures for data collection; and to ensure the feasibility of the questionnaire and study procedures in order to make adjustments, where necessary. Also, it was done to ensure that question formats and sequences were appropriate for the learners’ cognitive and reading ability. The pilot study included 25 secondary school learners from two secondary schools (one quintile 1 and one quintile 3) in Buffalo City Metropolitan Municipality. The learners of the pilot study and their schools did not form part of the main study.

### 2.10. Data Analyses

The final questionnaires and measurements of 1360 participants data were analyzed using both quantitative and qualitative data analysis techniques.

Weight and height measurements were used to calculate the body mass index (BMI; weight(kg)/height(m)^2^, which was used to classify learners as either underweight, normal weight, overweight, or obese based on internationally accepted cut-off values [22]. For learners younger than 18 years, age and sex specific cut-off values were calculated using the extended international International Obesity Task Force (IOTF) BMI cut-off values [23], while equivalent adult cut-off values were used for learners 18 years or older [23].

Quantitative data on the school food and nutrition environment, nutrition knowledge and anthropometric status were analyzed using descriptive statistics and are summarized as frequencies and percentages. Relationships between categorical data were determined using the Rao–Scott Chi-square test and the independent *t*-test was used to test for differences between two continuous variables. The complex survey design was incorporated in the calculations and a *p*-value of < 0.05 was considered statistically significant. All statistical analyses were performed with the Statistical Package for Social Science (SPSS) version 21.0 for windows (SPSS Inc., Chicago, IL, USA).

Qualitative data was analyzed using the word cloud technique [24]. The word cloud generator grouped common responses to emphasize recurring themes. Open ended written data from LO teachers and learners were read and re-read by the researcher, the supervisor, and another researcher in the field. As the researcher read through the collected data, any impressions were identified and recorded. The analysis was focused by looking at how the participants had responded to each question. This helped in identifying consistencies or differences. Ninety coherent categories were identified by reading and re-reading. Categories were developed until no new themes could be identified.

### 2.11. Ethics

Ethical approval for the study was granted by the Research Ethics Committee of the University of Fort Hare (Reference number: JIN011SOKE01). Permission to carry out the study was granted by the Departments of Basic Education and Department of Health, Eastern Cape Province, South Africa. Permission was requested from school principals to collect data from the learners and teachers within the school setting. Dates and time for data collection were arranged through consultations with the schools. Written consent was obtained from learners older than 18 years old, from parents/guardians of younger learners, and from LO teachers. In addition, written assent was obtained from all learners younger than 18 years before data collection.

## 3. Results

### 3.1. Demographic Data

Table 1 shows the demographic characteristics of the learners. Of the 1360 learners included in the study, 38.9% were male and 61.1% were female. In total, 163 (12%) of the learners were 20 years or older, with most (*n* = 101) of these older learners attending rural schools. There were 612 (45%) learners in quintiles 1 and 2 and 748 (55%) in quintile 3. Most learners were Black Africans (96.8%). There was an even number of learners per grade. In terms of parents’ education, 6.5% of mothers had no education while 16.2% had a primary school education. In fathers this was 11.1% and 15.6%, respectively. Furthermore, 6.8% of mothers had a high school and 16.2% a tertiary education compared with 51.1% and 21.1% of fathers, respectively. There were significant differences between quintile 1 and 2 schools versus quintile 3 schools for mothers’ (*p* = 0.003) and fathers’ (*p* = 0.006) education levels with a tendency for more mothers and fathers in quintile 3 schools to have a high school and tertiary education. There were significant urban–rural differences for race (*p* < 0.001) and fathers’ education level (*p* = 0.002). In the latter there were more fathers in urban areas having a high school or tertiary education.

### 3.2. Anthropometric Data

Figure 3 depicts the anthropometric status of the learners. The prevalence of underweight was 13.3% in males and 5.5% in females, normal weight was 76.8% in males and 58.4% in females, overweight was 7.4% in males and 23.8% in females, and obesity was 2.5% in males and 12.3% in females. Overweight/obesity was more prevalent in females compared to males (36.1% versus 9.9%; *p* < 0.001).

### 3.3. Meals Supplied by the National School Nutrition Program

Information on the meals supplied by the NSNP as reported by the learners is presented in Table 2. It was noted that 96% of learners received meals at school and that 85% received meals every day of the school week. Less than 6% of learners received a vegetable daily, furthermore the vegetables were generally added to the main dish and not provided separately. Most learners (95%) only received fruit once or twice a week.

The foods observed on the NSNP menu and those served on the day of the survey included dry beans, maize porridge and sour milk, cabbage, carrots, chicken, apples, green pea stew, rice, samp (maize kernels), and tinned fish. Some urban rural differences were noted with more rural schools providing meals (*p* < 0.001), on more days (*p* = 0.005), and using more fresh vegetables (*p* < 0.001). There was a significant difference in the type of fruit served with more fresh fruit at quintile 1 and 2 schools and more dry fruit at quintile 3 schools (*p* = 0.002).

### 3.4. Vendors, School Stores, and Lunch Boxes

Sources from where the learners obtained food are shown in Table 3. Most learners (73%) ate before school on the day of the survey. Only 12% brought a lunch box to school. Most learners (71%) brought money to school on the day of the survey, and the amount varied between Rand (R)1.00–30.00 (0.02 to 2 USD at exchange rate R18.00 to 1 USD). Learners spent a mean and standard deviation (SD) of R6.38 (4.23; 0.35 (0.24) USD) per day at school (data not shown). Foods were bought mostly from food vendors (59%).

The most popular food items bought by learners on the day of the visit were: crisps/chips (47.8%), fat cakes (39.6%), chocolates/sweets (27.1%), processed meats e.g., polony, sausages (12.5%), chicken heads, feet, organs (11.9%), fruit (10.7%), and bread (11.1%; Table 4). For those learners who brought a lunch box to school (*n* = 167), the main foods items in the lunch box were bread/sandwiches (64.7%), processed meats e.g., viennas, polony (23.4%), cold drinks (18.6%), and fruit (18%).

### 3.5. Learners Comments on the PSNP, Vendors, and School Stores

The learners expressed mixed views about foods provided by the NSNP and those sold in the school stores and by vendors as indicated in Table 5.

### 3.6. Vegetable Gardens

Only three schools (19%) had a vegetable garden. Forty-eight percent of all learners indicated that the purpose of a school vegetable garden was ‘learning about healthy eating’, while 27% said that the purpose of a school garden was ‘to produce food’.

Positive reasons about working in school vegetable garden were:“Through working in the school vegetable garden participants learn to plant and get fresh vegetables”; “it was fun”; “it made learners fit and it was a means of relaxation.”

Negative feelings about working in the school vegetable garden were:“No reasons for their feeling”; “they were lazy”; “they liked to help but did not have time”; “vegetables were seasonal and working in the garden coincided with learners’ times for studying’’; “Working in the garden was not fun and it was difficult”; “working in the garden gave learners skin irritations”; The “learners did not get any vegetables”; “the gardeners were selfish”; “Gardens are too small and learners did not usually help.”

### 3.7. Nutrition Knowledge of Learners

Questions that were poorly answered by more than 60% of learners, included the number of fruit and vegetable portions required daily, food to eat when overweight, foods containing fiber, starchy foods as basis of meals, amount of daily water required, food safety, dairy requirements, food items fortified with iodine, and importance of legumes. The majority of learners could correctly answer questions on eating sugar, salt, vitamin A sources, soya as a source of protein, and washing vegetables before consumption.

There was a significant difference in nutrition knowledge between the grades (*p* < 0.001; Table 6). There was an increase in good knowledge scores from grade 8 (5.2% to 26.6%) in grade 12. Those in the poor category decreased from 52.0% in grade 8 to 23.4% in grade 12. The difference between males and females was also significant (*p* = 0.002). In females 12.3% had a good score while this was 11.5% in males. However, 34.8% of females had a poor score compared with 45.9% in males. Those with a mother having a tertiary education scored highest in the good category (20.9%). When the mother had no education only 10.2% learners fell in the good category compared with 47.7% in the poor category. The difference in knowledge associated with mother’s education was significant (*p* < 0.002). The results for the father’s level of education showed a similar trend but was not significant (*p* = 0.571). There were no significant differences between the school quintiles (*p* = 0.801) or between school locations (*p* = 0.513).

### 3.8. Nutrition Knowledge of Life Orientation Teachers

Of the 18 LO teachers interviewed, 78.6% had received formal education training but only 28.6% received formal nutrition training. The most common form of training of those who had received training on nutrition was from the Department of Education. Other forms of training also included workshops, self-study through reading books, and as a prerequisite for a formal degree to become a teacher.

Of the 18 LO teachers interviewed, 5 (28%) did not know about the FBDGs and what they represent. The recommendations that the LO teachers believed to be stipulated in the FBDGs were: “Knowledge about dietary needs,” “eat more fruits and vegetables,” “balanced, healthy meals daily,” “eat salt, fat, and sugar sparingly,” “include fiber in your diet,” and “eat fresh food” in that order. Of the FBDGs only two were correct (Appendix A contains the FBDGs).

All LO Teachers agreed that they played an important role in promoting health and nutrition in schools. Some examples of common quotes from the teachers on the role that they should play in promoting health and nutrition included:“Teaching learners about healthy foods, how to prepare their foods, and about not using too much oil.”

Teaching learners about the basics of nutrition and food preparation was emphasized repeatedly among LO Teachers, with quotes including:“Learners must [be taught to eat] healthy food cooked in clean cooking areas, and to use clean utensils.”“Educate learners about basic food groups and their importance, sources of nutrients and required amounts.”“Teaching learners about the dangers of not eating healthily also featured strongly in the LO.”“Teaching learners healthy hygiene practices and how to keep their body in shape; avoid obesity, high blood pressure, and diabetes.”“Encouraging learners to eat healthy, such as eating vegetables and fruit and a balanced meal, and about maintaining a balanced diet to enhance quality living.”“Encourage learners to help in the green gardens where fruits are grown at school, and to have a fruit and vegetables garden at home.”“Helping learners to get the eating plan and correct foods, encourage their parents to give them adequate diet, encourage learners to eat breakfast, and to monitor their food intake, especially [to ensure an intake] of nutritious foods.”

In practice, the majority of LO teachers, 12 (85.7%), mentioned that they taught nutrition as part of their life orientation curriculum. The largest proportion of the nutrition curriculum was made up of teaching on healthy eating and maintaining a balanced diet, which was covered weekly throughout the school year. This was followed by a proportion of time (every two weeks) spent teaching on food groups, physical education/exercise and physical fitness, and on healthy foods. The least amount of time in the curriculum was dedicated to teaching on ill health and harmful substances in food production.

FBDGs that were not mentioned by the LO teachers were the following:Importance of the quality of fat eaten.Importance of drinking lots of clean, safe water.Making starchy foods the basis of most meals.Chicken, fish, meat, or eggs can be eaten dailyEating dry beans, peas, lentils, and soy regularly.Enjoying a variety of food.

It was clear from the answers provided that the LO teachers did not have a good knowledge of the FBDGs despite them being part of the LO curriculum.

### 3.9. The Integrated School Health Policy

While 73.9% learners indicated that health screening was done at their schools by a school nurse only 40.1% learners indicated that this included nutritional assessment such as anthropometry (Table 7). On site services were available according to 61.3% of learners, however individual counselling was only reported by 40.3%. Health promotion took place according to 90.1% learners with 67.6% reported that nutrition education was done. Rural schools did more health promotion (*p* < 0.002) and nutrition education than urban schools (*p* < 0.001). There were no significant differences between quintile 1 and 2 schools and quintile 3 schools regarding the health services provided.

## 4. Discussion

This study was done in the Eastern Cape province, which is one of the poorest provinces in the country. The sample consisted of quintile 1–3 schools in urban and rural areas and was therefore focused towards poorly resourced schools that are targeted by the NSNP. Most of the learners received a school meal daily. Foods bought at school were mostly unhealthy options. Few learners brought a lunch box to school which contained mostly sandwiches, processed meats, and cold drinks. Nutrition knowledge of the learners was generally poor and there were indications that the LO teachers did not know the FBDGs.

Age range within grades varied extensively and 12% of the learners were 20 years or older. Variation in age could be due to a number of reasons: including poverty, individual’s health, pregnancy, migration between areas leading to no provision for mid-term entry or lack of places for children, lack of appropriate documentation required to enroll in school (i.e., birth certificate), and lack of family support and encouragement [25].

From the demographic data it is clear that the majority of learners come from disadvantaged homes with many having fathers and/or mothers with little formal education. Despite this, 36.1% of females and 9.9% of males were overweight or obese. This figure is in line with those of three national surveys in South Africa [8,26,27] and emphasizes the importance of making the school environment as healthy as possible. The high prevalence of overweight/obesity in females is of particular concern because obesity generally tracks into adulthood and the consequences of maternal obesity during pregnancy result in poor outcomes for the mother and fetus [28]. For the mother, risks include gestational diabetes and preeclampsia while the fetus is at risk of congenital abnormalities and stillbirth. Obesity during pregnancy also carries additional risks in later life such as hypertension and heart disease in women and children having a higher risk of becoming obese in future.

In this study, the percentage of learners who were classified as thin (underweight) was low but highest in males. This trend has also been observed in national surveys in South Africa [8,26,27].

The disparity between obesity in males and females is difficult to explain. There has been an increase in obesity in South African adolescents between 2002 and 2008 and 2012 [26,27]. This increase is substantially higher in females particularly from mid-to-late adolescence [29]. A higher prevalence of obesity in females has also been documented in low- and middle-income countries [8,30]. Data suggests that adolescence is a critical risk period for fat deposition in girls. Another factor to consider in the disparity between the sexes is the low activity levels found in girls in urban settings and at older ages [31,32]. Furthermore, data suggest differences in the types of sedentary behavior adopted by males and females as another reason [32].

The coverage of the NSNP appears to be high in secondary schools in the Eastern Cape province, with more than 95% of the school learners receiving a meal every school day of the week. At least half of the learners received a vegetable 3–5 times a week as part of the school meal. Mostly, the vegetable was added to the main dish which could have meant long cooking times and loss of certain micronutrients. However, fruit was not as commonly served with the school meal; nearly 90% of learners only received a fruit once or twice a week. Similar findings were reported by Faber et al. who examined the NSNP at 90 purposively selected poorly resourced schools in South Africa [33]. The study found that schools did not comply with the mandate of serving vegetables and/or fruits everyday as stipulated in the 2011–2012 conditional grant 173 frameworks [34]. It is thus recommended that the Department of Basic Education find ways to enforce this mandate. Considering the overall low intake of vegetables and fruit in South African children [35] and the importance of eating sufficient amounts thereof to reduce the risk for developing certain non-communicable diseases and micronutrient deficiencies [36], the school meal together with good quality nutrition education can be a vehicle for increasing the intake of fruit and vegetables in learners.

Another South African study, which was undertaken in Bloemfontein in 10 randomly selected quintile 1–3 schools, analyzed samples of the meals eaten and found for the 11–18-year-old age group, the calcium standard (≥350 mg) was not met by any of the meals analyzed, and vitamin A and E levels were undetected [37]. This may have been due to the low intake and lack of variety in vegetables, with the main source of vegetables being cabbage. According to the Department of Basic Education the following should be served: “protein (soya, fish, eggs, milk, sour milk, beans, and lentils), fresh fruit and vegetable, and carbohydrate/starch. A variety of protein is served per week. Soya should not be served more than twice a week. Fats/oil, salt, and flavorings are added to make the meals tasty. Fresh vegetables or fruit should be served daily.” [16]. The results of the present study also demonstrated that the school meals did not comply with the guidelines of the FBDGs, which recommend the consumption of plenty of fruits and vegetables every day in order to meet nutrient requirements for protection against the development of non-communicable diseases [9].

Seventy-three percent of learners in the present study indicated that they had eaten before coming to school, leaving nearly 30% who did not. A study on breakfast habits in seven secondary schools in the North West province, South Africa found that 81% had breakfast with only 19% skipping it [38]. The latter is similar to a study by Faber et al. [33] who found that 22% of grade 5–7 learners did not eat breakfast.

In the present study more than 70% of learners brought money to school on the day of the visit and only 12% brought a lunch box. In Western Cape, South Africa, it was found that learners aged 10 to 12 years in 16 schools who carried a lunch box to school appeared to have greater dietary diversity, consumed more regular meals, had a higher standard of living, and greater nutritional self-efficacy compared with those who did not carry a lunch box to school [39].

Most food purchases at schools in the present study were made from vendors, either on or outside the school grounds. The main items purchased were unhealthy options such as crisps or chips, fat cakes, and chocolate or candies. These food items are generally high in fat, sugar, and salt, and are energy-dense; consumption thereof should be avoided. Various other studies in South Africa have reported similar results [33,40,41]. Learners in South African schools are therefore exposed to unhealthy food items within the school environment, which in the long term may encourage unhealthy eating habits which may in turn contribute to weight gain [42]. An in-depth survey on foods sold by food vendors in schools included in the current study showed that the unhealthier food options were cheaper sources of energy compared to healthier food options [43]. A study undertaken on school-aged adolescents in China found that BMI was positively associated with school environment factors such as availability of soft drinks at school and fast food outlets in school area [44]. Regular provision of fruit and vegetables should be included in the school meals. Food items sold to learners through vendors and school stores should be regulated and learners should be encouraged to carry a healthy lunch box.

A large percentage of learners knew that eating of foods low in sugar and salt are healthiest (results from the knowledge test), but this knowledge was not reflected in their choices of food items bought at school which were frequently high in sugar or salt. Healthier food options are less commonly sold by school food vendors, which may restrict learners’ food choices [33]. Nutrition knowledge of the learners was generally low, which is consistent with findings of other South African studies [45,46]. The LO teachers are involved in the implementation of the nutrition component the ISHP, yet few had formal training in nutrition education. In addition, results of this study show that their knowledge on the guidelines of the FBDGs was limited. Less than one third of LO teachers had received training on nutrition which was provided by workshops, self-study, through reading books, and as a prerequisite for a formal degree to become a teacher. The nutrition education component of the NSNP was strengthened by the Department of Basic Education providing an educator’s manual [12] on nutrition education comprising the FBDGs guidelines. The finding that the majority of LO teachers did not know the FBDGs implies that this manual is either not received or else not studied by the teachers. All LO teachers should be provided with specific guidelines to teaching nutrition as well as classroom materials to ensure that the information in the nutrition manual is clearly integrated into the classroom curriculum. LO teachers and school health nurses should have formal nutrition training.

With regard to evaluation of the ISNP, a large percentage of learners indicated that a school health nurse did health screening at their schools but only four out of ten indicated that this included nutritional assessment. Nutritional assessment is very important since the school nurse is able to establish whether under or overnutrition is present and to act accordingly by providing nutrition counselling/education. Hence such screening should be done at all schools regularly. On site services were available according to more than 60% of learners, however individual counselling was only reported by one in four. Significantly more appeared to be done at quintiles 1 and 2 than at quintile 3 schools and at rural compared with urban schools. While both the NSNP and the ISHP have good policies in place, the implementation seems to be lacking in many schools and the Department of Basic Education should improve the monitoring and evaluation of these policies. This is particularly important regarding which food items are sold on the school premises. Learners need to have access to healthy foods.

The nutrition education and promotion should also be extended to include parents and the school community. Pearson et al. [47] indicate that the family environment is an important influence on the dietary behaviors of learners and adolescents. Thus, family structure should be considered when designing programs to promote healthy eating behaviors.

### Limitations of the Study

This study was only undertaken in one province, so it is not possible to generalize to other provinces. A formal written nutrition knowledge test was not taken by LO teachers and many of their comments were subjective. Because so few schools had vegetable gardens it is difficult to make concrete conclusions or recommendations, but the lack of school gardens is certainly an aspect which the Department of Basic Education should investigate in order to fulfil the aim of the ISHP which promotes the establishment of school gardens.

## 5. Conclusions

The findings indicated that the nutritional practices of learners, and food items sold in school stores and vendors were unhealthy for the growth and development of the learners. The learners had poor nutrition knowledge, and many LO teachers who were responsible for teaching nutrition had no formal nutrition education training. Additionally, on-site services for nutritional assessment were absent, and the nutritional component, was not adequately implemented as stipulated in the ISHP. There is need for context-specific interventions by stakeholders to identify nutrition-related problems as early as possible. The provision of nutrition education to the learners and LO teachers would enhance their knowledge and perceptions about healthy eating practices. The ISHP should have clear guidelines on food items served and sold at schools in this setting. Unhealthy foods should not be tolerated on school premises.

## Figures and Tables

**Figure 1 ijerph-17-04038-f001:**
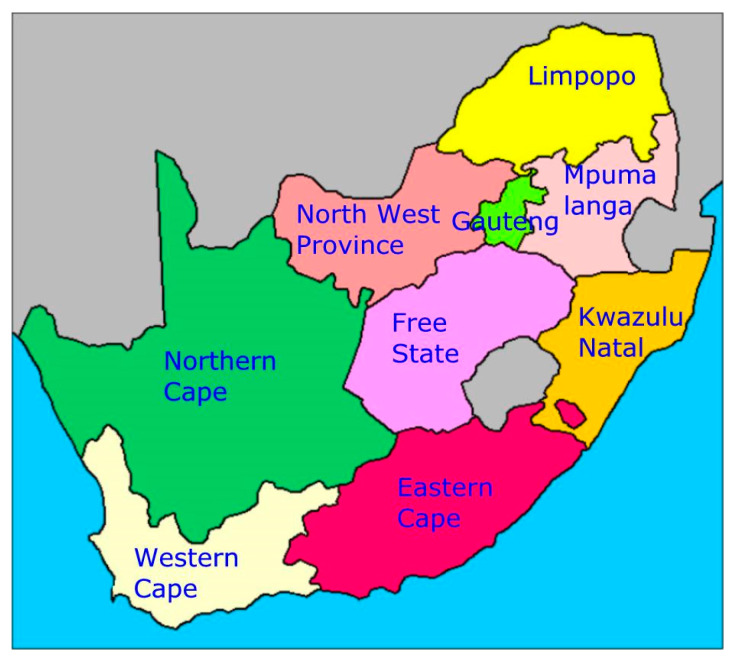
Map of South Africa showing the Eastern Cape province.

**Figure 2 ijerph-17-04038-f002:**
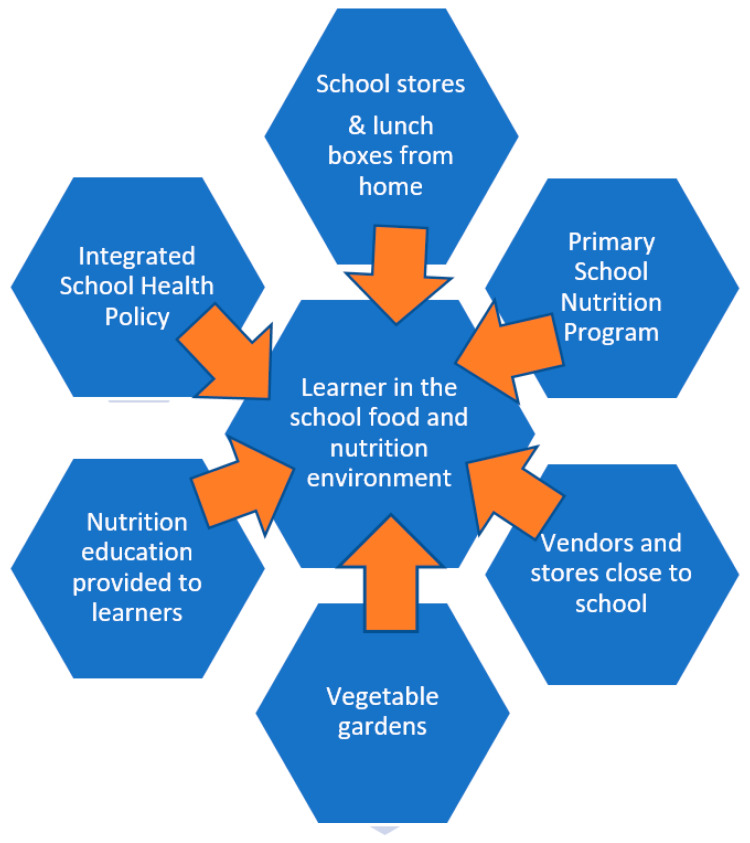
A schematic view of the factors influencing the school food and nutrition environment.

**Figure 3 ijerph-17-04038-f003:**
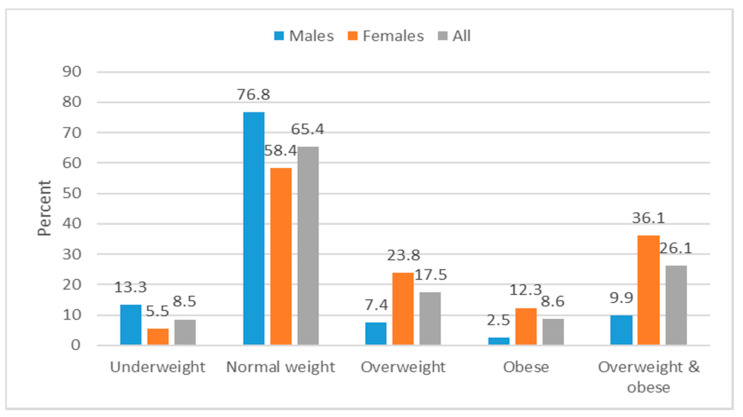
Anthropometric status of secondary school learners (*N* = 1360) in Eastern Cape schools.

**Table 1 ijerph-17-04038-t001:** Demographic characteristics of the learners by school quintiles and geographic areas.

Variables		All	Q 1 and 2	Q 3	Chi-Sq	Urban	Rural	Chi-Sq
		*n* (%)	*n* (%)	*n* (%)	*p*-Value	*n* (%)	*n* (%)	*p*-Value
Sample size		*N* = 1360	*n* = 612 (45%)	*n* = 748 (55%)		*n* = 653 (48%)	*n* = 707 (52%)	
Age (years)	11 to 16	593 (43.7)	244 (39.9)	349 (46.8)	0.380	312 (48.0)	281 (39.7)	0.224
	17 to 26	764 (56.3)	368 (60.1)	396 (53.2)		338 (52.0)	426 (60.3)	
Gender	Male	529 (38.9)	250 (40.8)	279 (37.3)	0.381	271 (41.5)	258 (36.5)	0.291
	Female	831 (61.1)	362 (59.2)	469 (62.7)		382 (58.5)	449 (63.5)	
Race	Black African	1315 (96.8)	596 (97.4)	719 (96.5)	0.695	617 (94.6)	698 (99.0)	<0.001 ***
	White	7 (0.5)	5 (0.8)	2 (0.3)		3 (0.5)	4 (0.6)	
	Mixed ancestry	35 (2.6)	11 (1.8)	24 (3.2)		32 (4.9)	3 (0.4)	
	Indian	2 (0.1)	-	-		-	-	
Grade	8	250 (18.6)	106 (17.3)	144 (19.6)	0.469	142 (22.2)	108 (15.3)	0.499
	9	188 (14.0)	63 (10.3)	125 (17.0)		98 (15.3)	90 (12.7)	
	10	336 (24.9)	138 (22.5)	198 (26.9)		161 (25.2)	175 (24.8)	
	11	381 (28.3)	180 (29.4)	201 (27.3)		130 (20.3)	251 (35.5)	
	12	192 (14.3)	125 (20.4)	67 (9.1)		109 (17.0)	83 (11.7)	
Mother’s highest education	None	88 (6.5)	57 (9.3)	31 (4.1)	0.003 **	33 (5.1)	55 (7.8)	0.152
	Primary	220 (16.2)	131 (21.4)	89 (11.9)		70 (10.7)	150 (21.2)	
	High	826 (60.8)	360 (58.8)	466 (62.4)		424 (65.0)	402 (56.9)	
	Tertiary	220 (16.2)	60 (9.8)	160 (21.4)		123 (18.9)	97 (13.7)	
	Don’t know	5 (0.4)	4 (0.7)	1 (0.1)		2 (0.3)	3 (0.4)	
Father’s highest education	None	150 (11.1)	87 (14.2)	63 (8.5)	0.005 **	50 (7.7)	100 (14.1)	0.002 **
	Primary	211 (15.6)	120 (19.6)	91 (12.3)		65 (10.1)	146 (20.7)	
	High	692 (51.1)	304 (49.7)	388 (52.4)		372 (57.6)	320 (45.3)	
	Tertiary	286 (21.1)	92 (15.0)	194 (26.2)		154 (23.8)	132 (18.7)	
	Don’t know	14 (1.0)	9 (1.5)	5 (0.7)		5 (0.8)	9 (1.3)	

*n* = number of learners; ** *p* < 0.01; *** *p* < 0.001; Q = Quintile; Chi-Sq = Rao–Scott Chi square.

**Table 2 ijerph-17-04038-t002:** Number of meals and foods received as part of the National School Nutrition Program.

		All	Q 1 and 2	Q 3	Chi-Sq	Urban	Rural	Chi-Sq
		*N* (%)	*n* (%)	*n* (%)	*p*-Value	*n* (%)	*n* (%)	*p*-Value
*Learners received NSNP meal*	Yes	1301 (96.0)	587 (95.9)	714 (96.1)	0.929	607 (93.7)	694 (98.2)	<0.001 ***
	No	54 (4.0)	25 (4.1)	29 (3.9)		41 (6.3)	13 (1.8)	
*Number of days of the week learners received the meal*	5 days	1143 (85.0)	518 (85.6)	625 (84.7)	0.677	505 (79.2)	638 (90.5)	0.005 **
	4 days	37 (2.8)	22 (3.6)	15 (2.0)		24 (3.8)	13 (1.8)	
	2–3 days	74 (5.5)	28 (4.6)	46 (6.2)		50 (7.8)	24 (3.4)	
	1 day	89 (6.6)	37 (6.1)	52 (7.0)		59 (9.2)	30 (4.3)	
*Number of days vegetables were served*	5 days	71 (5.8)	31 (5.1)	40 (6.3)	0.145	36 (6.4)	35 (5.2)	0.527
	3–4 days	726 (58.9)	402 (66.7)	324 (51.4)		352 (62.9)	374 (55.6)	
	1–2 days	436 (35.4)	170 (28.2)	266 (42.2)		172 (30.7)	264 (39.2)	
*Type of vegetables served*	Fresh	983 (76.0)	488 (80.1)	495 (72.3)	0.268	408 (66.2)	575 (84.8)	<0.001 ***
	Frozen	56 (4.3)	29 (4.8)	27 (3.9)		19 (3.1)	37 (5.5)	
	Dehydrated	30 (2.3)	7 (1.1)	23 (3.4)		21 (3.4)	9 (1.3)	
	Don’t know	225 (17.4)	85 (14.0)	140 (20.4)		168 (27.3)	57 (8.4)	
*How vegetables are served*	Separately	353 (27.5)	173 (28.6)	180 (26.6)	0.665	180 (29.6)	173 (25.7)	0.439
	Added to main dish	929 (72.5)	432 (71.4)	497 (73.4)		428 (70.4)	501 (74.3)	
*Number of days fruits are served*	5 days	17 (1.7)	0 (0.0)	17 (4.0)	&	16 (4.0)	1 (0.2)	<0.001 ***
	3–4 days	35 (3.6)	5 (0.9)	30 (7.1)		31 (7.7)	4 (0.7)	
	1–2 days	928 (94.7)	552 (99.1)	376 (88.9)		358 (88.4)	570 (99.1)	
*Type of fruit served*	Fresh	975 (94.8)	541 (96.6)	434 (92.5)	0.002 **	423 (93.6)	552 (95.7)	0.417
	Dry	54 (5.2)	19 (3.4)	35 (7.5)		29 (6.4)	25 (4.3)	
*How fruits are served*	Separately	971 (94.4)	533 (95.3)	438 (93.2)	0.414	422 (93.2)	549 (95.3)	0.453
	Added to main dish	58 (5.6)	26 (4.7)	32 (6.8)		31 (6.8)	27 (4.7)	

& = Cannot calculate Chi-square value if one of the cell values is zero. ** *p* < 0.01; *** *p* < 0.0001. Q = quintile; Chi-Sq = Rao–Scott Chi-square.

**Table 3 ijerph-17-04038-t003:** Dietary practices made in the school environment by grade 8–12 learners on the day the school was visited.

	Total	Responses	
	*n* *	Yes *n* (%)	No *n* (%)
Ate before school	1355	985 (72.7)	370 (27.3)
Brought lunch to school	1357	167 (12.3)	1190 (87.7)
Brought money to school	1352	954 (70.6)	398 (29.4)
Bought from school store	1283	186 (14.5)	1097 (85.5)
Bought from vendors inside school	1287	477 (37.1)	810 (62.9)
Bought from vendors outside school	1279	279 (21.8)	1000 (78.2)
Bought from café/spaza shop **	1279	124 (9.7)	1155 (90.3)
Bought from fellow learners	1277	132 (10.3)	114 (5.7)

*n* * differs because of missing values; ** Spaza shop is an informal shop selling only a few items and often not in a structured building.

**Table 4 ijerph-17-04038-t004:** Foods bought from school stores, vendors and brought from home by grade 8–12 learners on the day of the visit.

Group	Food Items Bought from School Stores and Vendors	*n* = 954	%
Starchy foods	Bread/sandwiches (11.1%), maize porridge (0.2%)	108	11.3
Protein-rich foods	Russians/sausages/viennas (3.6%), polonies (8.9%), pies (4.4%), chicken feet/heads/liver (11.9%), meat (0.3%), burgers (0.1%), peanuts (0.1%), fish (1.7%)	296	31.0
Fruit and vegetables	Vegetables (0.5%), fruit (10.7%)	107	11.2
Dairy products	Yoghurt (0.4%), cheese (0.2%)	6	0.6
Drinks	Cold drinks (4.6%), fruit juice (1.0%)	54	5.6
Confectionary	Biscuits (6.5%), fat cakes * (39.6%), pancakes (0.2%), scones (0.1%), wafers (0.3%), muffins (0.7%)	453	47.5
Candies	Chocolates or sweets (27.1%), lollies (1.3%)	271	28.4
Snacks	Crisps/chips (47.8%), popcorn (1.2%)	467	49.0
**Group**	**Food Items in Lunch Boxes**	*n* = 167	%
Starchy foods	Bread (64.7%), samp (maize kernels; 4.8%), cereal (1.8%), noodles (0.6%), rice (4.2%), stiff maize porridge (pap; 1.8%)	134	80.2
Protein-rich foods	Russian/viennas/ham (5.4%), eggs (7.2%), burger (4.2%), bacon (0.6%), polony (18.0%), meat (4.2%), chicken feet (0.6%), fish finger (1.2%)	69	41.3
Fruit and vegetables	Fruit (18.0%), potatoes (0.6%)	31	18.6
Dairy products	Yoghurt (1.2%), cheese (6.0%)	12	7.2
Drinks	Cold drinks (18.6%), tea (6.6%)	4	25.2
Confectionary	Fat cakes (3.0%), cake (2.4%)	9	5.4
Candies	Sweets (1.8%)	3	1.8
Snacks	Crisps (4.2%)	7	4.2
Spreads	Spreads (6.6%), peanut butter (1.2%)	13	7.8

Only 167 said “yes” to bringing a lunch box from home; 954 learners said “yes” to “Did you bring money to buy food at school today.” * balls of bread dough fried in oil.

**Table 5 ijerph-17-04038-t005:** Learners’ views on foods provided by the National School Nutrition Program (NSNP) and those sold in the school stores and by vendors.

	Positive Comments by at Least 10% of Learners	Negative Comments by at Least 10% of Learners
NSNP foods	“Food is good” (31.3%)	“Food being of poor quality” (11.8%)
	“Food is healthy” (31.3%)	“Small quantities” (11.7%)
	“Food is fresh” (27.1%)	
	“Food is delicious” (27.1%)	
	“Foods were well cooked “(10.2%)	
Food stores *	“Foods served were clean, fresh, good, delicious, healthy and always available. Foods bought at the tuck shop were cheap, with a fast response from attendants and they have varieties of food items”	“The food is unhealthy (too much oil), poor quality, without taste, rarely served vegetables, small in quantity, poorly cooked, stale, expired, unhygienic, irregularly and untimely served”
Vendors	“Food is tasty” (14.1%)	“Food is unhealthy” (19.6%)
	“Delicious” (14.0%)	
	“The food is always available” (12.6%)	

* Comments were by less than 10% of learners.

**Table 6 ijerph-17-04038-t006:** Nutrition knowledge of learners by demographic variables.

Variables	Good Score ≥ 24	Fair 17–23	Poor Score ≤ 16	Rao–Scott Chi-Square
	*n* (%)	*n* (%)	*n* (%)	*p*-Value
*Grades*				
8	13 (5.2)	107 (42.8)	130 (52.0)	*p* < 0.001 ***
9	13 (6.9)	89 (47.3)	86 (45.7)	
10	42 (12.5)	160 (47.6)	134 (39.9)	
11	42 (11.0)	210 (55.1)	129 (33.9)	
12	51 (26.6)	96 (50.0)	45 (23.4)	
*Gender*				
Male	61 (11.5)	225 (42.5)	243 (45.9)	*p* = 0.002 **
Female	102 (12.3)	440 (53.0)	289 (34.8)	
*Mothers’ highest level of education*				
None	9 (10.2)	37 (42.1)	42 (47.7)	*p* = 0.002 **
Primary school	26 (11.8)	101 (45.9)	93 (42.3)	(*p*-value
High school	82 (9.9)	417 (50.5)	327 (39.6)	calculated
Tertiary	46 (20.9)	108 (49.1)	66 (30.0)	without
Don’t know	0 (0.0)	1 (20.0)	4 (80.0)	DNO
*Fathers’ highest level of education*				
None	14 (9.3)	71 (47.3)	65 (43.3)	*p* = 0.571
Primary	20 (9.5)	103 (48.8)	88 (41.7)	
High school	84 (12.1)	341 (49.3)	267 (38.6)	
Tertiary	42 (14.7)	142 (49.7)	102 (35.7)	
Don’t know	2 (14.3)	4 (28.6)	8 (57.1)	
*Age (years)*				
11 to 16	61 (10.3)	284 (47.9)	248 (41.8)	*p* = 0.276
17 to 26	101 (13.2)	381 (49.9)	282 (36.9)	
*Quintiles*				
Quintiles 1& 2	64 (10.5)	309 (50.5)	239 (39.1)	*p* = 0.810
Quintile 3	99 (13.2)	356 (47.6)	293 (39.2)	
*School location*				
Urban	88 (13.5)	294 (45.0)	271 (41.5)	*p* = 0.513
Rural	75 (10.6)	371 (52.5)	261 (36.9)	

** *p* < 0.01; *** *p* < 0.001; DNO = Don’t know option.

**Table 7 ijerph-17-04038-t007:** Implementation of key health services packages at schools, according to the learners.

Services	No. Learners	Quintile 1 and 2	Quintile 3	Rao–Scott Chi-Square	Urban	Rural	Rao–Scott Chi-Square
	*n* (%)	*n* (%)	*n* (%)	*p*-Value	*n* (%)	*n* (%)	*p*-Value
Health screening	996 (73.6)	453 (74.5)	543 (72.8)	0.8138	498 (77.0)	498 (70.4)	0.2971
*Nutrition assessment*	540 (40.1)	251 (41.1)	289 (39.3)	0.7389	275 (43.0)	265 (37.5)	0.2515
On-site services	832 (61.3)	391 (64.1)	441 (59.0)	0.2602	395 (60.8)	437 (61.8)	0.8171
*Individual counselling*	543 (40.3)	241 (39.5)	302 (40.9)	0.7355	251 (39.2)	292 (41.3)	0.5895
Health promotion	1223 (90.1)	557 (91.3)	666 (89.0)	0.5281	563 (86.5)	660 (93.4)	0.002 **
*Nutrition education*	912 (67.6)	429 (70.3)	483 (65.4)	0.4009	384 (59.8)	528 (74.7)	*p* < 0.001 ***

Italics represent specific services coupled to nutrition activities. ** significant at *p* < 0.01; *** significant at *p* < 0.001.

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
