# Peer review of "The Food and Nutrition Environment at Secondary Schools in the Eastern Cape, South Africa as Reported by Learners"

_ijerph, 2020, doi:10.3390/ijerph17114038_

Round 1

Reviewer 1 Report

This is an interesting study. Perhaps the authors should mention about the dangers of obesity in pregnancy and the offspring. 

My only concern is that the paper is rather long and would benefit from being written more concisely.

Author Response

Reviewer 1 Comments

How comment was addressed

This is an interesting study. Perhaps the authors should mention about the dangers of obesity in pregnancy and the offspring. 

Thanks for important comments

Perhaps the reviewer missed the lines 408-413:

The high prevalence of overweight/obesity in females is of concern because obesity generally tracks into adulthood and the consequences of maternal obesity during pregnancy result in poor outcomes for the mother and fetus [26]. For the mother, risks include gestational diabetes and pre-eclampsia while the fetus is at risk of congenital abnormalities and stillbirth. Obesity during pregnancy also carries additional risks in later life such as hypertension and heart disease in women and children having a higher risk of becoming obese in future.

My only concern is that the paper is rather long and would benefit from being written more concisely.

We have removed Table 5 (from the original paper) and its results to shorten article. We have also included the subjective data on the foods provided by the National School Nutrition Program (NSNP) and sold in school stores and by food vendors as new Table 5 in order to reduce the content

Reviewer 2 Report

This is a well-presented manuscript.  And the findings should be useful for various stakeholders with interest in improving nutritional status among South African learners.

The only suggestion I have is to include more discussion on the gender disparity in obesity.  What information does/can the study provide to shed light on the large difference in overweight/obesity rate between male and female learners, despite that female learners generally have better knowledge about nutrition?  The insight should help elevate the utility and pertinence of this paper. 

Author Response

Reviewer 2 Comments

This is a well-presented manuscript.  And the findings should be useful for various stakeholders with interest in improving nutritional status among South African learners.

The only suggestion I have is to include more discussion on the gender disparity in obesity.  What information does/can the study provide to shed light on the large difference in overweight/obesity rate between male and female learners, despite that female learners generally have better knowledge about nutrition?  The insight should help elevate the utility and pertinence of this paper. 

Thank you for valuable comments

This paragraph has been added to lines 416 -423:

The disparity between obesity in males and females is difficult to explain. There has been an increase in obesity in South African adolescents between 2002 and 2008 and 2012 [26, 27]. This increase is substantially higher in females particularly from mid- to-late adolescence [29]. A higher prevalence of obesity in females has also been documented in low-and middle-income countries [8, 30]. Data suggests that adolescence is a critical risk period for fat deposition in girls. Another factor to consider in the disparity between the sexes is the low activity levels found in girls in urban settings and at older ages [31,32]. Furthermore, data suggest differences in the types of sedentary behavior adopted by males and females as another reason [32].

Reviewer 3 Report

The manuscript by Okeyo et al evaluate the food and nutrition environment at secondary schools in the Eastern Cape Province. The authors observe that 13.3% of males and 5.5% of females were underweight, while 9.9% of males and 36.1% of females were overweight or obese. The main food items purchased at school were fat cakes, chocolates, candies, and crisps/chips. The authors observed that there is need for context-specific interventions by stakeholders to identify nutrition-related problems. The study is comprehensive and generally supports the authors' conclusions. These findings may benefit from some additional clarification, as detailed below.

Comments

- The abstract should be better divided into introduction, material and methods, results and conclusions.

- These data should be better explained and studied considering the peculiar socio-economic condition. Furthermore, the results should be compared to data obtained from participants with different demographic characteristics.   

- The impact of the study should be better clarified and detailed.

- The manuscript should be edited to correct contextual and layout errors.

Author Response

https://susy.mdpi.com/user/manuscripts/resubmit/fff74273844738f90c33ede4a462370a

Reviewer 3 Comments

The manuscript by Okeyo et al evaluate the food and nutrition environment at secondary schools in the Eastern Cape Province. The authors observe that 13.3% of males and 5.5% of females were underweight, while 9.9% of males and 36.1% of females were overweight or obese. The main food items purchased at school were fat cakes, chocolates, candies, and crisps/chips. The authors observed that there is need for context-specific interventions by stakeholders to identify nutrition-related problems. The study is comprehensive and generally supports the authors' conclusions. These findings may benefit from some additional clarification, as detailed below.

Thank you for very valuable comments

- The abstract should be better divided into introduction, material and methods, results and conclusions.

With regard to the abstract the journal format does not include the headings suggested although we have written it as if they were present. There is an introduction, methods, results and conclusion

These data should be better explained and studied considering the peculiar socio-economic condition.

We added the following in lines 104-109:

. The quintile system is a ranking structure used by the Government of South Africa which groups schools according to the poverty level of the community where the school is located. Schools in quintile 1 and 2 represent the poorest and all the school funds come from the government. For the purpose of this study, schools from quintile 1 and 2 were grouped together while quintile 3 formed another group representing a slightly higher socio-economic status, with the highest being quintile 5.  All schools in quintiles 1-3 receive a government provided meal daily.

 Also data better explained in terms of SES (quintiles)

Table 1 and lines 237-238and lines 242-245 :

There were 612 (45%) learners in quintiles 1 & 2 and 748 (55%) in quintile 3.

There were significant differences between quintile 1 & 2 schools versus quintile 3 schools for mothers’ (p=0.003) and fathers’(p=0.006) education levels with a tendency for more mothers and fathers in quintile 3 schools to have a high school and tertiary education.

Table 2 lines 269-270:

There was a significant difference in the type of fruit served with more fresh fruit at quintile 1 and 2 schools and more dry fruit at quintile 3 schools (p=0.002).

Table 6 lines 331-332:

There were no significant differences between the school quintiles (p=0.810) or between school location (p=0.513).

Table 7 lines 391-392:

There were no significant differences between quintile 1 and 2 schools and quintile 3 schools regarding the health services provided.

Furthermore, the results should be compared to data obtained from participants with different demographic characteristics.   

The focus of the study was on the most disadvantaged schools and urban rural differences. Results were obtained from different quintiles and urban rural settings. In terms of ethnicity the numbers were too small to do statistical comparisons and it was not our focus.

The impact of the study should be better clarified and detailed.

The aim of the study was hence to evaluate the food and nutrition environment within secondary schools in urban and rural areas in the Eastern Cape in order to assess how healthy this environment was for learners.

The impact (aim) was:

The findings indicated that the nutritional practices of learners, and food items sold in school stores and vendors were unhealthy for the growth and development of the learners. The learners had poor nutrition knowledge, and many LO teachers who were responsible for teaching nutrition had no formal nutrition education training. Additionally, on-site services for nutritional assessment were absent; and the nutritional component, not adequately implemented as stipulated in the ISHP. There is need for context-specific interventions by stakeholders to identify nutrition-related problems as early as possible. The provision of nutrition education to the learners and LO teachers would enhance their knowledge and perceptions about healthy eating practices. The ISHP should have clear guidelines on food items served and sold at schools in this setting. Unhealthy foods should not be tolerated on school premises.

- The manuscript should be edited to correct contextual and layout errors.

Manuscript has been edited again

Reviewer 4 Report

How Obesogenic is the School Food and Nutrition Environment at Secondary Schools in the Eastern Cape Province of South Africa?

Overall comments
The focus of the study seems to be a bit unclear. At the end of the “Introduction” section, the authors appear to position the study as an evaluation of countrywide education policy in South Africa related to nutrition and diet, but the text seems to lose that focus frequently. For instance, the authors say that they study breakfast eaten before school (but not at school, if I’ve understood correctly) and box lunches brought from home as part of the school nutrition environment. This seems outside the scope of the school’s jurisdiction unless the federal nutritional policy dictates that schools must assess what students eat outside of school and what they bring into school from home. Elsewhere, the text seems to focus on adolescent overweight and obesity. The title suggests that the paper will evaluate how obesogenic the school environment is (which while it seems that the school environment is obesogenic, I think it is harder to establish the degree of “obesogenicity” of the school food environment, which seems to be what the title promises). I think it would help the paper’s readability and contribution if the authors determined exactly what they want to paper to be about, make the title about that message, and then make sure to focus on that point. After reading the paper, it seems that the common theme revolves around policy (the ISHP and/or FBDGs). I would recommend the authors focus on this.

Abstract
General: The title suggests that the paper is going to be about the food and nutrition environment, but the data and tests reported in the abstract are all focused on individual behavior or knowledge rather than the environment. This makes the concluding sentence of the abstract feel unsupported by evidence.

Page 1, Line 21 (P1L21): I feel a stronger opening sentence would highlight the growing concern about overweight and obesity world-wide. If it’s already known that adolescent females in South Africa experience high levels of overweight/obesity, the contribution of the study is diminished—the authors would in that case just be corroborating what is already known from other studies. Is there anything else novel about this research that the authors could highlight as a contribution (instead of just reframing the opening sentence to connect the study to global concerns about high rates of overweight/obesity)?

P1L26: What are fat cakes? Could these food items just be generalized to “unhealthy foods” or some other all-encompassing term, at least for the abstract? Did these food purchasing patterns differ by sex?

P1L29: This p-value is highly specific (and raises questions about whether the p-value is really less than 0.0157 or if that was the precise p-value for that test). The authors could reduce the number of digits they report after the decimal point in my opinion.

P1L31: “Also, the majority of teachers had no formal nutrition training and their responses to knowledge questions were poor and indicated that they were not familiar with the South African food-based dietary guidelines despite the availability of such a manual for nutrition educators.” Were the teachers who answered these questions nutrition educators? If they were not specifically nutrition educators, were these manuals also available to them?

Introduction
P2L41: Could the authors think of a more engaging way to introduce the paper? What if a researcher who does not do research in or know much about overweight and obesity were to read the first sentence or two of papers to decide what to read? The facts stated in the first sentence are meaningful to people who can bring their own context to what they read about overweight/obesity, but may be relatively meaningless for those unfamiliar with this topic.

P2L48: “In addition, obesity tracks from adolescence to adulthood.” This seems like an important fact that could motivate the study in general.

P2L54: This may be picky, but while I agree that malnutrition may deprive people of the ability to enjoy life […], I don’t see how it can deprive them of the right to enjoy life […]. Rights are legal; malnutrition is not (though it may result from societies’ decisions about what people’s rights are with respect to food access, social support, etc.).

P2L65: Is the “its” in “…practices of learners because of its emphasis on multilevel linkages…” referring to socio-ecological models? If so, it should be plural.

P2L69: “…learners spend on average 24 hours a week in a school environment…” Is this averaged across the whole year, or just during the school year? What constitutes the school year in South Africa?

P2L76: “Other aims of the ISHP include addressing health barriers to learning in order to improve education outcomes of accessibility to schooling, school retention and school achievements.” This sentence as written is slightly hard to process (for me). I propose deleting “…to learning in order to improve education…” so that the sentence would read, “Other aims of the ISHP include addressing health barriers to improve schooling accessibility, school retention, and school achievement.”

P2L88: An empirical evaluation wouldn’t be investigated—it would be conducted (or something similar).

P6L177: How were the point cut-offs determined for the nutrition knowledge scores?

P6L186: There’s an “R” missing in the word “three.”

P6L211: “Open ended written data from LO teachers and learners were read and re-read.” To what end—what were the researchers doing while they read and re-read these open-ended responses? Is this when they formed the “impressions” mentioned in the next sentence? Does “read and re-read” mean that one researcher read each open-ended response two times? Or was it as many times as it took them to form an impression?

P7L232: “In terms education” should probably be something like “In terms of parents’ education”.

Author Response

Reviewer 4

Overall comments
The focus of the study seems to be a bit unclear. At the end of the “Introduction” section, the authors appear to position the study as an evaluation of countrywide education policy in South Africa related to nutrition and diet, but the text seems to lose that focus frequently. For instance, the authors say that they study breakfast eaten before school (but not at school, if I’ve understood correctly) and box lunches brought from home as part of the school nutrition environment. This seems outside the scope of the school’s jurisdiction unless the federal nutritional policy dictates that schools must assess what students eat outside of school and what they bring into school from home. Elsewhere, the text seems to focus on adolescent overweight and obesity. The title suggests that the paper will evaluate how obesogenic the school environment is (which while it seems that the school environment is obesogenic, I think it is harder to establish the degree of “obesogenicity” of the school food environment, which seems to be what the title promises). I think it would help the paper’s readability and contribution if the authors determined exactly what they want to paper to be about, make the title about that message, and then make sure to focus on that point. After reading the paper, it seems that the common theme revolves around policy (the ISHP and/or FBDGs). I would recommend the authors focus on this.

Thank you very much. Your comments were very insightful and relevant and we have done our best to follow your suggestions.

Numerous studies have indicated that many learners do not eat breakfast before coming to school and was one of the drivers of the government school meal program. While not strictly in the school environment we were interested to see whether breakfast habits had changed since the school meals had been introduced. For this reason we also included the lunch boxes since the Education Department nutrition curriculum focuses on healthy choices in lunch boxes even if this is not a compulsory policy. If children do not bring a healthy lunch box they are compelled to buy what is available in the school environment which is generally high in sugar and fats and energy-dense.

We have changed the title in order to deal with the comment regarding degree of obesogenicity: The food and nutrition environment at secondary schools in the Eastern Cape as reported by learners

Abstract
General: The title suggests that the paper is going to be about the food and nutrition environment, but the data and tests reported in the abstract are all focused on individual behavior or knowledge rather than the environment. This makes the concluding sentence of the abstract feel unsupported by evidence.

Abstract has been reworded

Abstract: Overweight and obesity are growing concerns in adolescents, particularly in females in South Africa. The aim of this study was to evaluate the food and nutrition environment in terms of government policy programs, nutrition education provided, and foods sold at secondary schools in the Eastern Cape Province. Sixteen schools and grade 8-12 learners (N=1360) were randomly selected from three health districts comprising poor disadvantaged communities. Based on age and sex specific BMI cut-off values, 13.3% of males and 5.5% of females were underweight, while 9.9% of males and 36.1% of females were overweight or obese. The main food items purchased at school were unhealthy energy-dense items such as fried flour dough balls, chocolates, candies, and crisps/chips. Nutrition knowledge scores based on the South African food-based dietary guidelines (FBDGs) were poor for 52% to 23.4% learners in Grades 8 to 12, respectively. Female learners generally had significantly higher nutrition knowledge scores compared to their male counterparts (p = 0.016). Questions poorly answered by more than 60% of learners, included the number of fruit and vegetable portions required daily, food to eat when overweight, foods containing fiber, and importance of legumes. It was noted that the majority of teachers who taught nutrition had no formal nutrition training and their responses to knowledge questions were poor indicating that they were not familiar with the FBDGs which are part of the curriculum. Nutrition assessment as part of the Integrated School Health Program was done on few learners. Overall however, despite some challenges the government national school meal program provided meals daily to 96% of learners. In general, the school food and nutrition environment were not conducive for promoting healthy eating.

Page 1, Line 21 (P1L21): I feel a stronger opening sentence would highlight the growing concern about overweight and obesity world-wide. If it’s already known that adolescent females in South Africa experience high levels of overweight/obesity, the contribution of the study is diminished—the authors would in that case just be corroborating what is already known from other studies. Is there anything else novel about this research that the authors could highlight as a contribution (instead of just reframing the opening sentence to connect the study to global concerns about high rates of overweight/obesity)?

Included an opening sentence about the growing concern of overweight and obesity globally in lines 43-50: “The number of obese children and adolescents (aged five to 19 years) worldwide has risen tenfold in the past four decades. If current trends continue, more children and adolescents will be obese than moderately or severely underweight by 2022. Combined, the number of obese five to 19-year-olds rose more than tenfold globally, from 11 million in 1975 to 124 million in 2016 [1]

Something else novel has been added line 97-101:

“  The aim of the study was hence to evaluate the food and nutrition environment within secondary schools in urban and rural areas in the Eastern Cape in order to assess how healthy this environment is for learners. The study is novel since it takes all aspects of the nutrition environment into consideration

P1L26: What are fat cakes? Could these food items just be generalized to “unhealthy foods” or some other all-encompassing term, at least for the abstract? Did these food purchasing patterns differ by sex?

We have added the following sentence lines 26-27:

“The main food items purchased at school were unhealthy energy-dense items such as fried dough balls, chocolates, candies, and crisps/chips.”

P1L29: This p-value is highly specific (and raises questions about whether the p-value is really less than 0.0157 or if that was the precise p-value for that test). The authors could reduce the number of digits they report after the decimal point in my opinion.

 The p value = 0.016 and has been corrected and all p values with less digits.

P1L31: “Also, the majority of teachers had no formal nutrition training and their responses to knowledge questions were poor and indicated that they were not familiar with the South African food-based dietary guidelines despite the availability of such a manual for nutrition educators.” Were the teachers who answered these questions nutrition educators? If they were not specifically nutrition educators, were these manuals also available to them?

 The following has been added to lines 65-69:

A special manual comprising information on the FBDGs was developed by the Department of Basic Education and made available to life orientation (LO) teachers at schools [10]. Life orientation is a subject which covers basic important issues regarding health such as nutrition, hygiene, and physical activity. The subject is taught by general teachers and not nutrition educators.

Could the authors think of a more engaging way to introduce the paper? What if a researcher who does not do research in or know much about overweight and obesity were to read the first sentence or two of papers to decide what to read? The facts stated in the first sentence are meaningful to people who can bring their own context to what they read about overweight/obesity, but may be relatively meaningless for those unfamiliar with this topic.

“In addition, obesity tracks from adolescence to adulthood.” This seems like an important fact that could motivate the study in general.

 The following section has been moved up to emphasize the tracking issue lines 46-50: The high prevalence of overweight /obesity in females is of concern due to the many health problems associated with overweight and obesity in young people. Overweight and obesity are risk factors for non-communicable diseases (NCDs) including cardiovascular diseases, type 2 diabetes, certain cancers and coronary heart disease [2]. In addition, obesity tracks from adolescence to adulthood [3]. Preventing adolescents’ obesity early in the life-course is therefore essential.

Also, most of line 58-72 has been revised

P2L54: This may be picky, but while I agree that malnutrition may deprive people of the ability to enjoy life […], I don’t see how it can deprive them of the right to enjoy life […]. Rights are legal; malnutrition is not (though it may result from societies’ decisions about what people’s rights are with respect to food access, social support, etc.).

We agree and have changed the sentence to lines 54-55:

“Malnutrition undermines economic growth, perpetuates poverty and can make children vulnerable to abuse and exploitation [5].”

Is the “its” in “…practices of learners because of its emphasis on multilevel linkages…” referring to socio-ecological models? If so, it should be plural.

Changed in lines 73-76:

Socio-ecological models provide a conceptual framework to study factors influencing the dietary practices of learners because of their emphasis on multilevel linkages, the relationships among the multiple factors that impact on health and nutrition and the focus on the connections between people and their environments [11].

P2L69: “…learners spend on average 24 hours a week in a school environment…” Is this averaged across the whole year, or just during the school year? What constitutes the school year in South Africa?

Added line 77-79:

For example, in South Africa, secondary learners spend on average 35  hours a week in a school environment over four terms a year with breaks of two to four weeks between terms.

Primary school children spend an average of 24 hours a week

: “Other aims of the ISHP include addressing health barriers to learning in order to improve education outcomes of accessibility to schooling, school retention and school achievements.” This sentence as written is slightly hard to process (for me). I propose deleting “…to learning in order to improve education…” so that the sentence would read, “Other aims of the ISHP include addressing health barriers to improve schooling accessibility, school retention, and school achievement.”

Lines 189-193

This whole section was reduced to comply with shortening the paper

The ISHP provides a policy framework for adequate school environments and includes three school health packages and services, namely: health assessment and screening; health education and promotion; and on-site services. The questionnaire completed by the learners included questions on whether they had received the services of each of the three health packages, namely health assessment and screening; health education and promotion; and on-site services by school nurses.

P2L88: An empirical evaluation wouldn’t be investigated—it would be conducted (or something similar).

Changed in lines 91-93:

However, an empirical evaluation of the ISHP to provide evidence for policymakers concerning its effectiveness and implementation, particularly as it pertains to the dietary behaviors and weight status of secondary school learners, has rarely been conducted.  

P6L177: How were the point cut-offs determined for the nutrition knowledge scores?

Lines 183-184: Cut-off points decided by research team

The level of nutrition knowledge scores of the learners were divided into the following scores<=16 (≤40%): poor; 17-<24 (40-≤63%); fair; >=24 (≥63%): good.

P6L186: There’s an “R” missing in the word “three.”

Line 212 corrected:

The questionnaire completed by the learners included questions on these services for each of the three health packages.

P6L211: “Open ended written data from LO teachers and learners were read and re-read.” To what end—what were the researchers doing while they read and re-read these open-ended responses? Is this when they formed the “impressions” mentioned in the next sentence? Does “read and re-read” mean that one researcher read each open-ended response two times? Or was it as many times as it took them to form an impression?

Yes agreed this has not been well described.

Line 238  made more explicit:

Open ended written data from LO teachers and learners were read and re-read by the researcher, the supervisor, and another researcher in the field.

P7L232: “In terms education” should probably be something like “In terms of parents’ education”.

Line 240 corrected:
In terms of parents’ education,….

Round 2

Reviewer 3 Report

No comments

Reviewer 4 Report

The authors have adequately addressed my comments.